: PLOS | ONE

# Detection and analysis of pulse waves during sleep via wrist-worn actigraphy

Johannes Zschocke[1,2], Maria Kluge[3], Luise Pelikan[3], Antonia Graf[3], Martin Glos[3], Alexander Müller[4], Rafael Mikolajczyk[1], Ronny P. Bartsch[5], Thomas Penzel[3], Jan W. Kantelhardt[2]*

1 Institute of Medical Epidemiology, Biostatistics and Informatics, Faculty of Medicine, Martin-Luther-University Halle-Wittenberg, Halle, Germany, 2 Institute of Physics, Martin-Luther-University Halle-Wittenberg, Halle, Germany, 3 Interdisziplinäres Schlafmedizinisches Zentrum, Charite - Universitätsmedizin Berlin, Berlin, Germany, 4 Klinik und Poliklinik für Innere Medizin I, Technische Universität München, Munich, Germany, 5 Department of Physics, Bar-Ilan University, Ramat Gan, Israel

* jan.kantelhardt@physik.uni-halle.de

**Data Availability Statement:** The data are available from the project "Detection and analysis of pulse waves via wrist-worn actigraphy during sleep" at OSFHOME (osf.io; location Frankfurt, Germany) under the DOI 10.17605/OSF.IO/XSBHW.

## Abstract

The high temporal and intensity resolution of modern accelerometers gives the opportunity of detecting even tiny body movements via motion-based sensors. In this paper, we demonstrate and evaluate an approach to identify pulse waves and heartbeats from acceleration data of the human wrist during sleep. Specifically, we have recorded simultaneously full-night polysomnography and 3d wrist actigraphy data of 363 subjects during one night in a clinical sleep laboratory. The acceleration data was segmented and cleaned, excluding body movements and separating episodes with different sleep positions. Then, we applied a bandpass filter and a Hilbert transform to uncover the pulse wave signal, which worked well for an average duration of 1.7 h per subject. We found that 81 percent of the detected pulse wave intervals could be correctly associated with the R peak intervals from independently recorded ECGs and obtained a median Pearson cross-correlation of 0.94. While the low-frequency components of both signals were practically identical, the high-frequency component of the pulse wave interval time series was increased, indicating a respiratory modulation of pulse transit times, probably as an additional contribution to respiratory sinus arrhythmia. Our approach could be used to obtain long-term nocturnal heartbeat interval time series and pulse wave signals from wrist-worn accelerometers without the need of recording ECG or photoplethysmography. This is particularly useful for an ambulatory monitoring of high-risk cardiac patients as well as for assessing cardiac dynamics in large cohort studies solely with accelerometer devices that are already used for activity tracking and sleep pattern analysis.

## 1 Introduction

Full-night polysomnography (PSG) has been regarded as the reference standard in sleep medicine since 1968 [1, 2]. Besides signals used for sleep stage classification, respiratory activity and an electrocardiogram (ECG) are usually recorded and analyzed [3]. However, the applicability

**Funding:** This study was supported by the German Israel Foundation (GIF, http://www.gif.org.il) grants I-1298-415.13/2015 (for JZ and JK) and I-1372-303.7/2016 (for MG, RB, and TP) and the German National Cohort study (www.nako.de), funded by the Federal Ministry of Education and Research (BMBF) and the Helmholtz Association (for JZ, AM and RM). JZ acknowledges support from a Minerva Short-Term Research Grant. We acknowledge the financial support within the funding programme Open Access Publishing by the German Research Foundation (DFG). The funders had no role in study design, data collection and analysis, decision to publish, or preparation of the manuscript.

**Competing interests:** The authors have declared that no competing interests exist.

of PSG for the assessment of sleep characteristics in large prospective studies is limited due to its costs and its intricacy, requiring many electrodes and cables attached to the subject's head and chest.

Alternatively to PSGs, actigraphy (or accelerometry) is commonly used to monitor human sleep/wake cycles [4–8]. Usually, the accelerometer is placed on the subjects' wrist of the non-dominant arm. Advantages of accelerometry are low costs, higher availability, easy recording of multiple nights, and a less disturbed natural sleep [9, 10]. However, its accuracy varies between different sleep variables and depends on population-specific characteristics [7, 9, 11]. Yet, recent technological progress has led to advanced recording devices with high temporal resolution (above 100 Hz), high acceleration resolution (down to 3 mg $\approx$ 0.03 m/s$^2$), and separate recording of all three spatial directions (see e.g., [12] for a review).

First investigations that demonstrated physiological relevance in the distribution and auto-correlations of wrist activity fluctuations independent of level of physical activity were published by Hu et al. [13, 14]. In later studies it has been shown that wrist activity fluctuations are also related to the circadian rhythm and to the role of the suprachiasmatic nucleus in the brain [15, 16] that is responsible for regulating many different body functions on a 24-hour cycle.

In this paper, we present an approach for exploiting nocturnal wrist accelerometry recordings to identify pulse waves and heartbeats, and assess detection accuracy of individual heartbeats. By comparing with simultaneously recorded ECGs (as part of clinical PSG), we demonstrate that accelerometry could help assessing sleep-related changes not only in heart rate but also in heart rate variability (HRV), including measures that rely on changes between neighboring inter-beat intervals. In Section 2, we summarize previous efforts to derive heart activity without electrodes. In Section 3, we describe our database and present the methods and data processing approaches. Section 4 reports our results including the achieved heartbeat detection reliability, statistics for pulse transit times, and influences of respiration on pulse wave intervals, including age dependences. We conclude in Section 5.

## 2 Alternative approaches for assessing heart activity

Besides the ECG as gold standard for heart rate and HRV measurements [17], there are several other methods to detect heartbeats not requiring electrodes attached to the body.

### Plethysmography

A common approach for measuring pulse waves is photoplethysmography. It relies on the propagation of pulse waves throughout the body. During heart contraction, blood is pumped into the arteries, creating a pressure ("pulse") wave. The velocity and shape of the pulse wave depend mainly on arterial stiffness that is affected by age, physical fitness, heart rate, body height, and gender [18]. According to O'Rourke et al. [18], the ideal aortic pulse wave profile is described as "sharp upstroke, straight rise to the first systolic peak, a definite sharp incisura, and near exponential pressure decay in late diastole". With the pulse waves' propagation to the periphery, the systolic pressure increases, while diastolic and mean pressures decrease due to increased arterial stiffness and incoming reflected pulse waves [19]. Pulse wave measurements on the wrist typically show wave profiles that are a superposition of three waves: an incident wave due to blood flow as well as two reflected waves from the hand and from the lower body, respectively [19, 20]. In plethysmography the pulse wave is recorded by light reflexion and light absorption [21]. We refer to [22] for an early application of using pulse wave intervals from plethysmography to study cardiac dynamics and investigate HRV during rest and exercise without ECG electrodes.

## Seismocardiography

Recording chest wall motion via radar-facilitated distance measurements is a possible but rather intricate approach [23]. Another not frequently used method is seismocardiography, where acceleration sensors placed on the chest wall measure the vibrations caused by heartbeats [24]. With higher resolution of acceleration sensors this technique became more interesting in the last years [25–27]. Seismocardiography is closely related to ballistocardiography, a method which measures whole body motions (or vibrations) caused by the heartbeat. Sensors are commonly placed on the chair or bed of the subject [26]. Seismocardiography and ballistocardiography are often used as synonyms.

Measurements of seismocardiography not only detect heartbeats but also respiratory activity. Beside respiration ($< 1$ Hz), low frequency (0.6 to 5 Hz) chest wall motions related with heart muscle contraction and high frequency ($> 5$ Hz) chest wall vibrations related with acoustic waves of the valve closing are measured [25, 28]. Both signals can be used to detect respiration and heartbeats [29].

## Accelerometry

In spectral analysis of nocturnal wrist-worn acceleration measurements also two distinct peaks have been identified [6]. As shown in Fig 1, there is a rather narrow peak at $\approx 0.3$ Hz reflecting respiratory activity and a much broader peak around 10 Hz, which we somewhat incorrectly coined "tremor peak" in the original publication. Both peaks are most pronounced if the variations of acceleration are at an intermediate level for the nocturnal recording, i.e., there is neither strong motion activity (often corresponding to wakefulness episodes or turns, Fig 1

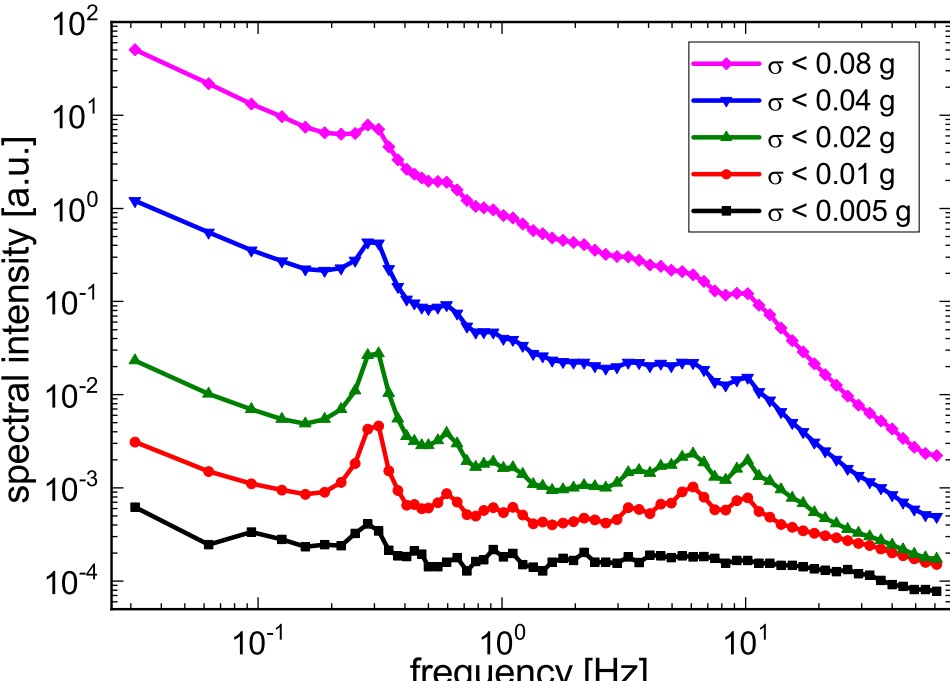

**Fig 1. Spectral intensity of wrist acceleration during different levels of motion activity.** Periodograms are shown for five exponentially increasing acceleration variance thresholds (black: smallest threshold; magenta: largest threshold). Peaks related with respiratory motion (at $\approx 0.3$ Hz) and pulse waves (at $\approx 6 - 10$ Hz) are most clearly visible for intermediate acceleration variance levels (red and green curves) (after [6]).

magenta curve), nor hardly any motion at all (probably corresponding to time intervals when the arm is practically fixed between other body parts and the bed; Fig 1 black curve). Hence, when exploiting these peaks, we cannot expect to get reliable data at all times, but only when there is an intermediate level of total acceleration variations.

Looking closer at the high-frequency ($\approx 10$ Hz) peak, we have recently identified the main reason for its broadness—the corresponding oscillation is strongly damped, being triggered approximately each second, but then decaying within $\approx 0.2$ s. By comparing with a simultaneously recorded ECG, the origin of this "tremor peak" finally became clear to us—it is caused by the pulse wave transversing the subject's wrist shortly after the heartbeat and probably triggering a short wiggling of the wrist and/or the recording device attached to it. These high frequency vibrations in the wrist caused by the arrival of the pulse wave have some analogy to the above mentioned high frequency chest wall vibrations ($> 5$ Hz) as detected by seismocardiography.

# 3 Materials and methods

## Measurements

All sleep recordings were performed at the clinical sleep laboratory of the Charité-Universitätsmedizin Berlin, Germany, between April 2017 and December 2018. The study was approved by the ethics committee of the Charité-Universitätsmedizin Berlin and registered at the German Clinical Trial Register (DRKS) with ID DRKS00016908. In total, 392 subjects were included and signed informed consent. During their first diagnostic night at the sleep laboratory, all subjects wore a SOMNOwatch™ plus device, recording simultaneously 3d wrist acceleration of the non-dominant arm at 128 Hz sampling rate and a one channel ECG at 256 Hz. Furthermore, full PSG (including electroencephalography (EEG), electrooculography (EOG), electromyography (EMG), ECG, respiratory effort, etc.) was recorded using either an ALICE, an Embla®, or a SOMNOscreen™ PSG system.

Due to noisy or low quality ECG recordings, 29 subjects were excluded from further analysis. The final 363 subjects (180 females, 183 males), aged between 18 and 80 years (mean $50.1 \pm 13.7$ years) with average body mass index $28.0 \pm 5.8$ kg/m$^2$, had an average time in bed (TiB) of $7.6 \pm 0.8$ h. All subjects were reffered to the sleep laboratory with complaints and an indication to test for sleep disorders. In Table 1 we list the frequency of sleep disorders classified by ICSD-3.

**Table 1. Overview of all subjects included in the analysis.** Subjects with multiple diagnoses are counted in each appropriate diagnosis line, i.e., multiple times. The last line reports data for all subjects irrespective of diagnosis. The column "duration" reports the median total duration (per subject) of all pulse wave intervals (PWI) correctly associated with corresponding heartbeat intervals from the ECG at an accuracy limit of 0.1 s (see Methods and also Table 2). It is followed by the median fraction of correctly associated PWI and the corresponding median Pearson cross correlation $r$ in the subsequent columns (see Results section for details).

| Diagnosis | females | males | duration | cor. PWI | $r$ |
|---|---|---|---|---|---|
| No sleep disorders | 19 | 18 | 1.2 h | 0.82 | 0.89 |
| Sleep-related breathing disorders | 67 | 115 | 1.2 h | 0.79 | 0.93 |
| Insomnia | 65 | 25 | 1.3 h | 0.84 | 0.93 |
| Central disorders of hypersomnolence | 36 | 20 | 1.3 h | 0.78 | 0.95 |
| Sleep-related movement disorder | 35 | 26 | 1.0 h | 0.80 | 0.94 |
| Parasomnias | 9 | 7 | 1.0 h | 0.74 | 0.95 |
| Circadian rhythm sleep-wake disorders | 1 | 9 | 1.7 h | 0.86 | 0.95 |
| Other sleep disorders | 8 | 7 | 1.3 h | 0.83 | 0.95 |
| All subjects | 180 | 183 | 1.3 h | 0.81 | 0.94 |

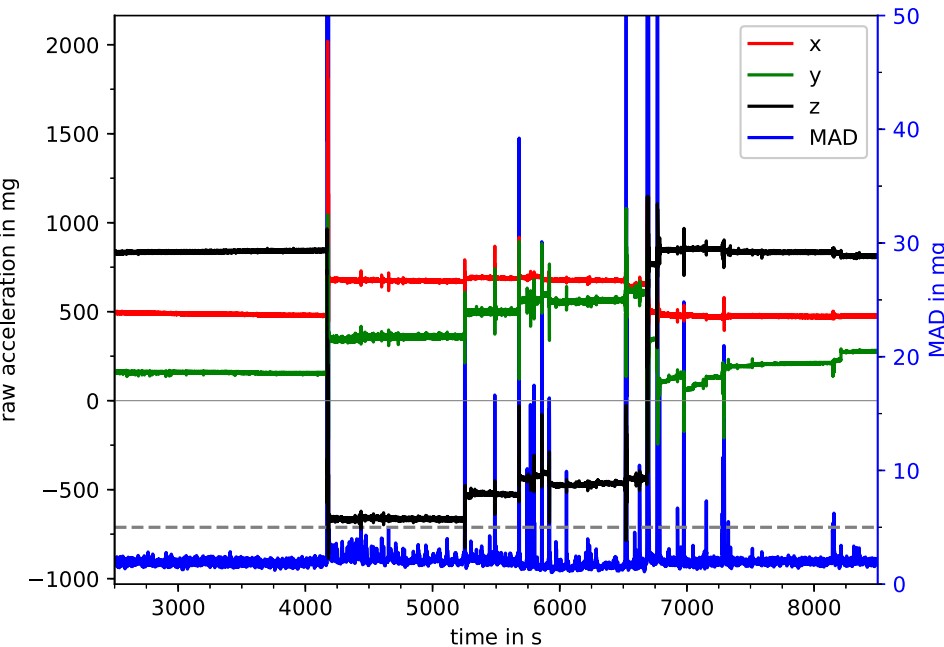

**Fig 2. Raw acceleration signals and MAD.** A typical nocturnal 100 minute part of the raw data is shown with all three directions ($\ddot{x}$—red, $\ddot{y}$—green, $\ddot{z}$—black) of the acceleration signal according to the left vertical axis. Furthermore, one-second mean amplitude deviation (MAD) values are plotted in blue with the MAD threshold of 5 mg shown as dashed gray line according to the right vertical axis. Time periods in which the $\text{MAD}_{1s}$ values are above this threshold were labeled as a position change (i.e., change in wrist orientation, see for example the peak at 4150 s).

## Data segmentation and stationarity transform

Fig 2 shows a section of a typical nocturnal recording of a three-axis wrist accelerometer. There are segments of nearly constant acceleration, e.g., from $t = 2500$ s to $t \approx 4100$ s, from $t \approx 4200$ s to $t \approx 5250$ s, etc. During such segments, the broad orientation of the wrist with respect to the gravitational field (i.e., the vertical direction) is constant, so that all three components of acceleration ($\ddot{x}$, $\ddot{y}$, and $\ddot{z}$) represent mainly the constant projections of the gravitational acceleration $g_0 = 9.81$ m/s$^2$ on each axis of the device. Specifically, the $x$ axis points towards the hand, while $y$ and $z$ are orthogonal to $x$ and to each other with directions possibly changing between the subjects and throughout the night.

Segments of nearly constant acceleration components (due to gravitational force only) are interrupted by obvious changes of the wrist orientation with respect to the gravitational field. In order to automatically identify such broad orientation changes, we calculated a mean ampli-tude deviation (MAD) very similar to $\text{MAD}_{5s}$ introduced by Vähä-Ypyä et al. [30],

$$\text{MAD}_{1s}(t) = \frac{1}{128} \sum_{i=t\times128\text{Hz}-63}^{t\times128\text{Hz}+64} |a_i - \langle a_i \rangle|, \tag{1}$$

with $a_i = \sqrt{\ddot{x}_i^2 + \ddot{y}_i^2 + \ddot{z}_i^2}$ and $\langle a_i \rangle = \frac{1}{128}\sum_{i=t\times128\text{Hz}-63}^{t\times128\text{Hz}+64} a_i$, considering non-overlapping windows of one second here. In our cleaning procedure, all acceleration data are set to zero, if their cor-responding $\text{MAD}_{1s}(t)$ values exceed the ad-hoc threshold of 5 mg (= 0.005 $g_0$).

In the following, we refer to continuous time segments not interrupted by $\text{MAD}_{1s}$ values above the 5 mg threshold as sleeping position segments (SPS). We assume that the subjects did not change their sleeping positions without increased motion activity. In each SPS and for each acceleration component ($\ddot{x}_i$, $\ddot{y}_i$, and $\ddot{z}_i$), we eliminated the offsets (caused by gravity) by

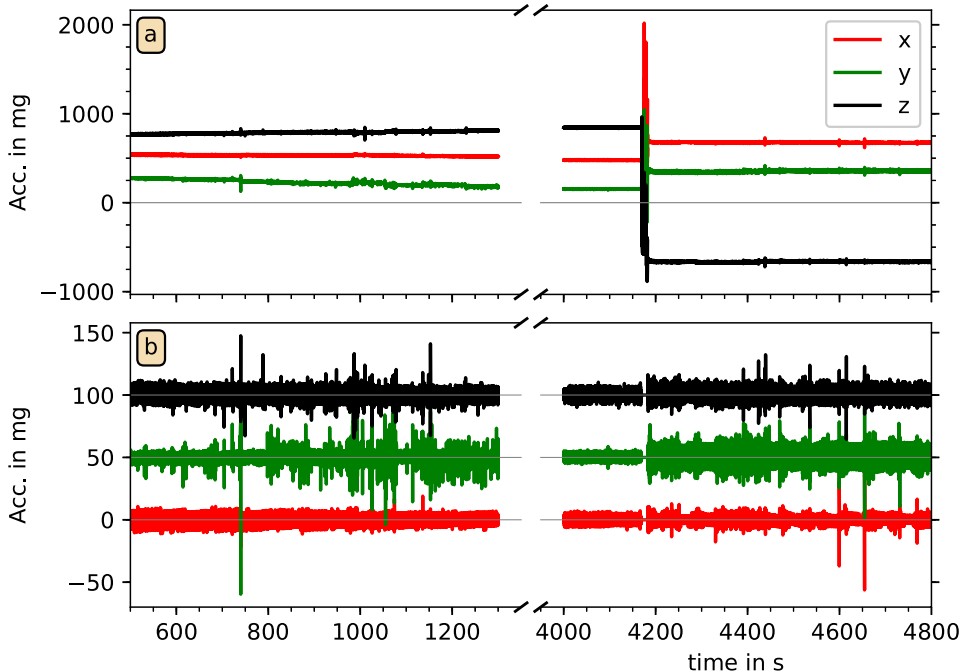

**Fig 3. Raw and cleaned acceleration data.** The upper panel shows raw acceleration data ($\ddot{x}$—red, $\ddot{y}$—green, $\ddot{z}$—black) during two parts of a recording. A weak steady trend appears in the $\ddot{y}$ component on the left hand side, and a wrist position change appears at time 4150 s on the right hand side (as already mentioned in Fig 2). The lower panel shows a magnification of the cleaned data in both parts, with constant offsets and slow trends removed. All acceleration data with their corresponding $MAD_{1s}$ values above the MAD threshold were set to zero. The data for $\ddot{y}$ and $\ddot{z}$ have been shifted upwards by multiples of 50 mg for visibility.

subtracting the mean values of each second. Fig 3 shows two examples for this acceleration data preprocessing procedure. We note that a subtraction of mean values of each second turned out to be sufficient, since—besides the stronger motions exceeding the threshold— there are only very slowly drifting wrist orientation changes with respect to the gravitational axis, see, e.g., Fig 2 in the range from $t = 6800$ s to 7200 s. The subtraction of one-second averages also turned out to be sufficient for an elimination of the slow ($\approx 0.3$ Hz) respiratory signal often superimposed on the acceleration recordings via tiny turns of the wrist, see also Fig 4(a) and 4(b). Approximately stationary acceleration data with zero means and only short interruptions have thus been obtained in the data cleaning procedure.

## Pulse wave peak (PWP) and pulse wave interval (PWI) detection

Fig 3 shows that amplitude variations of typically $10 - 40$ mg remain after the acceleration data have been cleaned. These signals often exhibit a rather periodic behavior, see Fig 4(a) and 4(b) for details at a high temporal resolution. Note that the corresponding variations of measured acceleration are quite small and in fact close to the resolution of the recording device, which digitizes measurements between $-6\,g_0$ and $+6\,g_0$ at 12 bits, yielding a resolution of 2.9 mg. The small spikes at an approximate periodicity somewhat below one second already look like indications of heartbeats. Most probably, pulse wave propagations through the wrist lead to tiny turns of the wrist with respect to the vertical (gravitational) axis, resulting in changes of the gravitation vector projections on the axes of the acceleration recording device.

In the next step, for a better identification of the pulse wave events, we applied a fast Fourier transform (FFT) based band pass filter with a lower cutoff at 5 Hz and an upper cutoff at 14 Hz

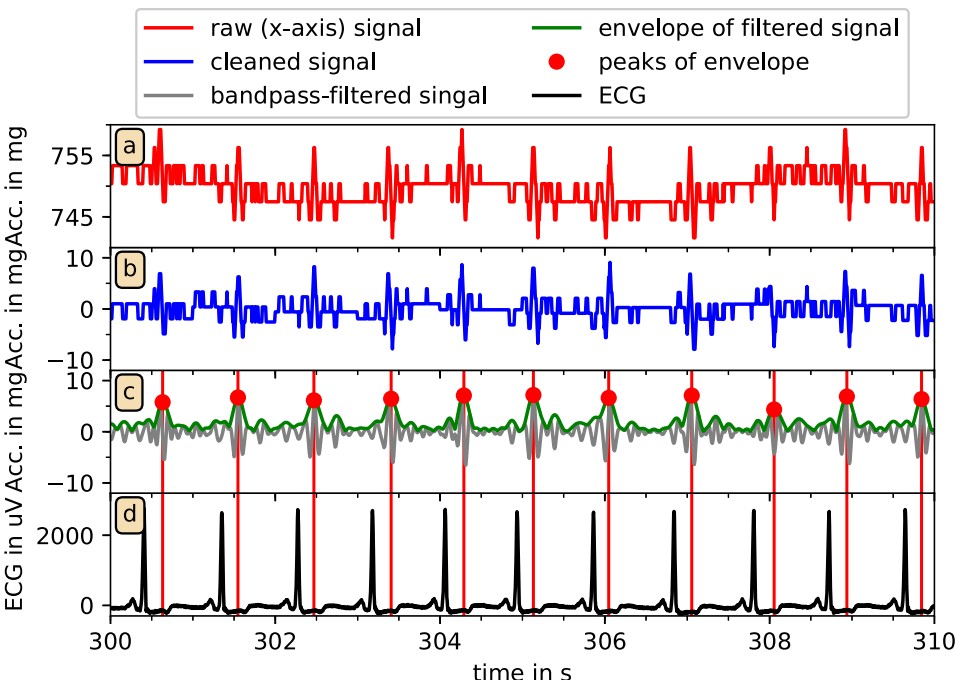

**Fig 4. Reconstruction of pulse waves from acceleration data.** In (a) and (b) the raw (red) and cleaned (blue) *x*-axis acceleration data from a typical recording is shown for ten seconds. Panel (c) shows the signal after the 5-14 Hz FFT-bandpass-filtering (gray), the absolute of the Hilbert transform (green), and the results of the peak detection (red dots). In (d) the ECG-signal (black) is presented and compared to the pulse wave peak (red vertical lines). Please also note the impact of respiration in panel (a) leading to a modulation of the acceleration data with a period of about 4 s. These modulations are removed in panel (b) by subtracting one-second averages.

to the data of each axis. We have empirically optimized these cutoff frequencies by studying acceleration data of many subjects. A typical result is shown in Fig 4(c), gray line. For a reliable identification of the pulse wave-related peaks we then applied a Hilbert transform [31] to the cleaned and band-pass filtered acceleration data $\tilde{a}_i$ of each axis to supplement the original signal with an imaginary part and calculate the instantaneous amplitudes $A(t)$ in an analytic signal approach,

$$\tilde{a}(t) + i\mathrm{HT}[\tilde{a}(t)] = A(t)\exp[i\varphi(t)], \tag{2}$$

(Fig 4(c), green line). Among the first applications of this approach to physiological dynamics are the works of Ivanov et al. who used Hilbert transform to detect the amplitude of heart rate variability fluctuations [32, 33].

Finally, a peak detection algorithm was used to identify candidate peaks in these pseudo pulse wave time series (Fig 4(c) red dots). Specifically, a local maximum of the time series was accepted as the next pulse wave peak candidate if it exceeded an ad-hoc threshold of 2.9 mg and has a minimum distance to the previous accepted peak of 0.5 s. Note that, in analogy to R peak detection from ECGs, we refer to the peaks as pulse wave peaks (PWP) and to the time intervals between them as pulse wave intervals (PWI). We also note that PWP are not real pulse (pressure) wave peaks, but closely related to them. Fig 4(d) shows that each detected PWP is clearly associated with an R peak of the simultaneously recorded ECG. The Figure also demonstrates the delay of the PWP with respect to the R peaks caused by pulse wave transit time (PTT) from the heart to the wrist.

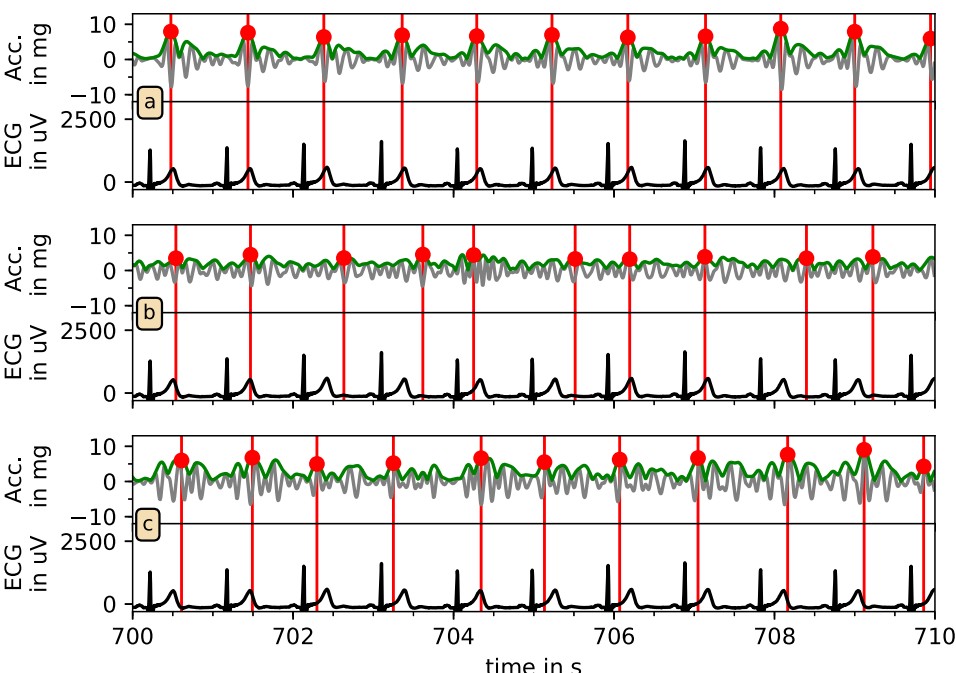

**Fig 5. Selection of best axis for pulse wave reconstruction.** The bandpass-filtered acceleration signals (gray), the corresponding Hilbert amplitudes (green) and the automatically identified candidate pulse wave peaks (red) are compared with the ECG (black) for all three axes of acceleration for another typical ten-second section of the recordings. Clearly, the detection reliability differs between the axes; in this case the best choice for beat detection is the *x* axis shown in panel (a).

Having three time series of candidate PWP (from each accelerometer axis) we have to select the best position estimate for each pulse wave. Fig 5 shows that—in this case—the *x* axis (part (a)) yields the most consistent PWP positions with respect to the R peaks in the ECG. However, in order to select candidate PWP without assessing ECG signals, we defined two criteria to choose—for each SPS—the best acceleration axis. Firstly, the plausibility of the candidate PWP was checked by calculating average pulse rate, requiring a value of at least 40 beats per minute for a plausible signal. If no signal was plausible, the considered SPS has not been used for further analysis. Secondly, if two or all three signals passed the first test, we applied a self-consistency check. Specifically, we calculated auto-correlation functions for the Hilbert amplitude signals and chose the axis with the highest auto-correlation peak in the range from 0.4 s to 1.5 s (40 beats/min to 150 beats/min).

In the final step, we calculated the PWI. In analogy with similar approaches for checking the validity of detected R peaks in an ECG, the duration of each PWI must either be between 0.7 s and 1.5 s (corresponding to instantaneous values of 40 beats/min to 86 beats/min) or in the range of ± 30 percent of the previous PWI. Furthermore, we accepted only uninterrupted sequences of at least 20 PWI, thereby excluding very short SPS. This way we obtained time series of PWI comparable to RR-interval (RRI) time series. For comparing the two types of time series, we have calculated mean heart rate and the following two standard parameters of HRV [17]: standard deviation of normal-normal intervals (SDNN) and root mean square of successive differences (RMSSD).

To derive RRI time series from ECG, these data were processed with the software LibRasch [34]. We visually verified and manually checked QRS classifications (normal, ventricular

**Table 2. PWI reconstruction correctness for different accuracy limits.** Results are shown for five different maximally accepted differences between RRI and PWI (accuracy limits). The column "time" reports the median total duration of all PWI that are correctly associated with RRI (per subject). As in Table 1, it is followed by the fraction of correctly associated PWI and the corresponding Pearson cross correlation $r$. Values are median [0.25 quantile; 0.75 quantile].

| accuracy limit | time | correct PWI | $r$ |
|---|---|---|---|
| 0.05 s | 1.2 [0.4; 2.1] h | 0.73 [0.61; 0.83] | 0.96 [0.93; 0.98] |
| 0.10 s | 1.3 [0.5; 2.3] h | 0.81 [0.69; 0.89] | 0.94 [0.88; 0.96] |
| 0.15 s | 1.4 [0.5; 2.4] h | 0.84 [0.74; 0.91] | 0.89 [0.82; 0.94] |
| 0.20 s | 1.4 [0.5; 2.4] h | 0.88 [0.77; 0.94] | 0.85 [0.77; 0.91] |
| 0.25 s | 1.4 [0.5; 2.4] h | 0.88 [0.77; 0.94] | 0.83 [0.74; 0.90] |

ectopic, and supra-ventricular ectopic) and corrected them if necessary. Noisy parts where no QRS detection was possible were manually marked and excluded from further analysis.

### Comparison of PWI and RRI

Due to the transition time between each heartbeat and the arrival of the pulse wave at the wrist, a direct comparison of R peaks and PWP is not appropriate. Hence, we compared RRI and PWI, defining their temporal positions as the middle of each interval. A PWI matches an RRI if its position is within 0.0 to 0.3 s after the RRI's position. Note that the empirical value of this threshold is post-hoc justified by the distribution of PTT we observe in Fig 9. A matching PWI is considered as correct, if its value is less than 0.1 s smaller or larger than the corresponding RRI (accuracy limit). This accuracy limit has been varied to check for its effects on the results (see Table 2 below).

## 4 Results and discussion

### Reliability of reconstructed pulse wave intervals (PWI)

As described in the method section, we have reconstructed PWI from wrist accelerometry time series independent of the ECG. In our 363 datasets we were able to reconstruct PWI during 25.7 percent of the total recording time (in the sleep laboratory), which corresponds to an average duration of 1.7 h per subject.

Fig 6 shows a direct comparison of tachograms of PWI and RRI derived from the simultaneous acceleration and ECG recordings of two subjects. A very close match between the two curves can be seen, although one ventricular heartbeat in (a) is not correctly identified by the pulse wave analysis, and there seems to be an increased high-frequency (HF) component in the PWI data.

In total, 80.9 percent of the detected PWI could be correctly associated with RRI at an accuracy limit of 0.1 s. In terms of time, 1.3 hours of correct PWI were detected per night. Table 2 reports median values and inter-quartile ranges regarding the achieved levels of correctness for the reconstructed PWI also for smaller and larger accuracy limits (see Methods subsection on Comparison of PWI and RRI above). We find that the results do not strongly depend on this accuracy limit, since the fraction of correctly reconstructed and associated PWI varies only between 0.73 and 0.88 for a broad variation of the limit from 0.05 s to 0.25 s (Table 2). In particular, increasing the limit from 0.2 s to 0.25 s does not change this fraction. Since most correctly detected PWI differ from the RRI by less than 0.05 s (1.2 h total time per subject) and doubling or tripling the limit increases this total time only by 0.1 h and 0.2 h, respectively, we conclude that an accuracy limit of 0.1 s is appropriate for a fair comparison.

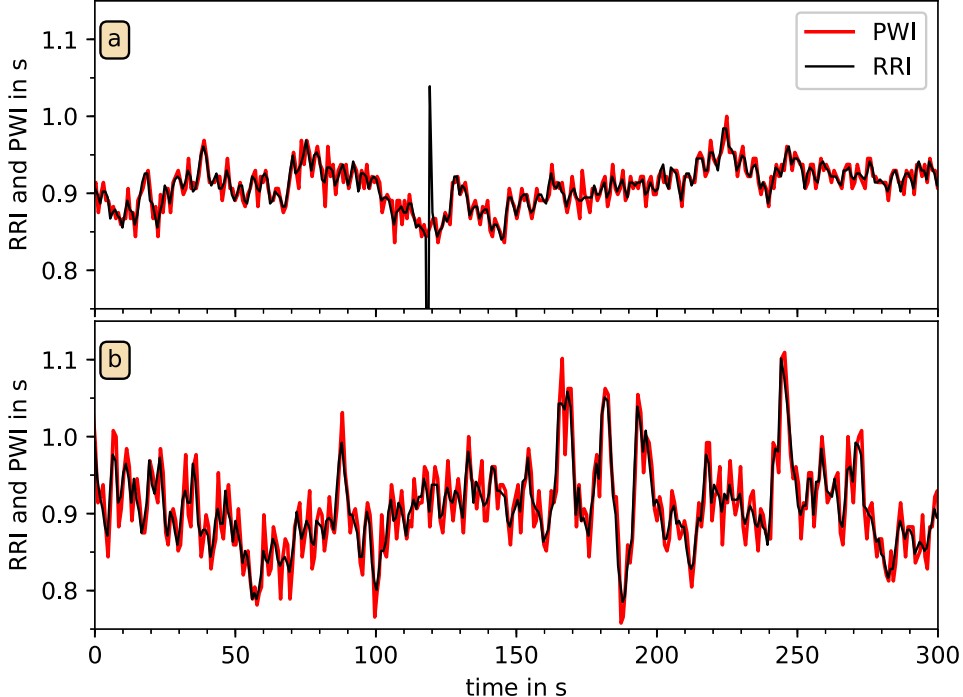

**Fig 6. Comparison of tachograms from RRI and PWI.** In these two examples from different subjects, RRI derived from the ECG (black) and PWI independently derived from wrist accelerometry (red) are plotted versus time. All detected PWP and all R peaks were used; the PWI are strongly correlated with RRI. However, unexpected heartbeat events, as for example the premature beat at $t$ = 120 s in (a), are not present in the PWI signal.

At the 0.1 s accuracy limit, the Pearson cross-correlation coefficient $r$ between the values of reconstructed PWI and correctly associated RRI is quite large, $r$ = 0.94. Note, that $r$ can only be calculated with respect to the PWI correctly associated with RRI. As expected, it decreases somewhat with larger accuracy limits as more and more PWI are included. However, $r$ = 0.85 at the 0.2 s limit is still very good. Note that $r$ is based on only 351 datasets (instead of 363), since no correct PWI were detected in 12 datasets. In addition, the different ICSD-3 diagnoses of the subjects have little effect upon our results as shown in the last two columns of Table 1.

Next we want to check the variation of the PWI detection performance of our algorithm across all 363 subjects. Fig 7 shows histograms for the total time of detected PWI in each subject and the fraction of correctly reconstructed and associated PWI. Although we have 74 datasets with less than 30 minutes of usable acceleration signals, most recordings—233 datasets—yield reconstructed PWI totaling between 30 minutes and 3.5 hours. In five datasets, we could detect PWI for more than 5.5 hours. The histogram for the fraction of correctly reconstructed and associated PWI (Fig 7(b)) has a small peak at 0 to 10 percent (15 datasets), which includes 12 recordings without any correctly detected PWI, and rises to a maximum at 80 to 90 percent correct detection. In 216 datasets more than 80 percent of the detected PWI were correct.

These percentages hardly depend on the age of the subjects. No systematic differences between three age groups of approximately equal size (see Table 3) can be observed when comparing the corresponding histograms for each color in Fig 7. This indicates that the reconstruction of pulse waves from wrist actigraphy as presented in this paper does not depend on age. Furthermore, the results in Table 3 show that there is no systematic age dependence in the PWI algorithm selection of particular orientation axes. Across all age groups, the $y$ axis

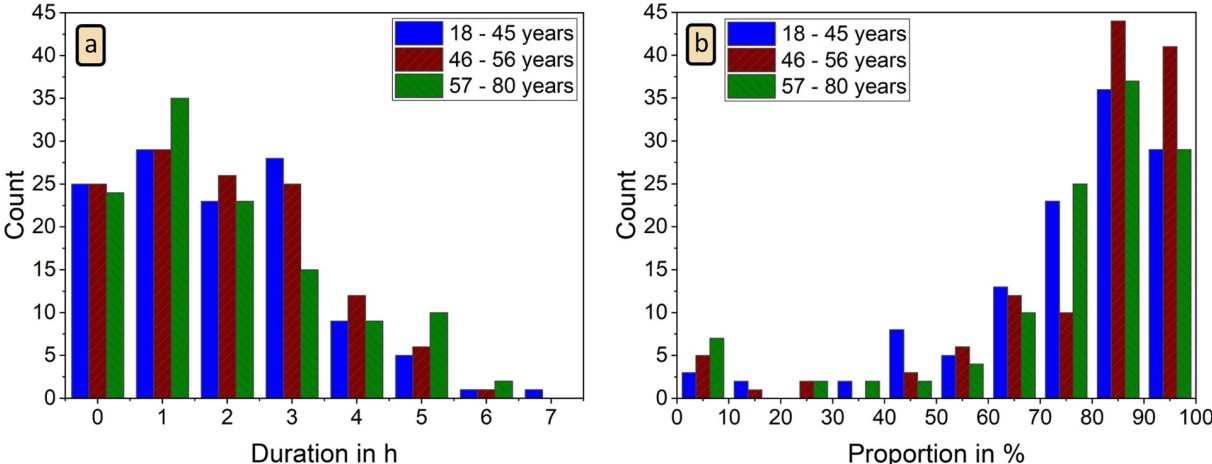

**Fig 7. Histograms for total PWI duration and correctness of PWI identification.** (a) Histogram of total duration of all detected PWI for each night; (b) histogram of fraction of correctly reconstructed and associated PWI (as compared to RRI from ECG) for each night. No clear differences between the three age groups (see Table 3) can be seen.

acceleration data is selected most frequently for the PWI detection, followed by the *z* axis data. The *x* axis, pointing towards the hand, is only quite rarely selected.

## Estimation of heart rate and HRV parameters from PWI

Table 4 compares heart rates and two standard HRV parameters [17] across all three steps of our PWI-RRI matching procedure. Clearly, the effect on resulting mean heart rate is minimal. In addition, there seems to be only little selection bias regarding correctly reconstructed and not reconstructed heartbeats, since the SDNN and RMSSD values for the matched RRI subset are close to the values for the whole (nocturnal) RRI time series. Furthermore, the results for

**Table 3. Age dependence of SDNN and RMSSD and origin of matched PWI.** For three similarly sized age groups the fractions of matched PWI derived from each of the three accelerometer axes are reported, showing the *y* axis data is used for more than half of all PWI correctly associated with RRI. The mean values of the HRV parameters SDNN and RMSSD and the mean PTT as derived from matched RRI and PWI are shown for comparison with literature [35, 39]. Regarding SDNN and RMSSD, all differences between the young age group and the other two groups are highly significant ($p \leq 0.002$), while no significant differences occur between the intermediate and the elderly group. The results indicate that the reduction in SDNN and RMSSD with age is similar in RRI (as derived from the ECG) and PWI (as reconstructed through wrist actigraphy). The differences between the mean PPT values of the young group and the other two groups are weak but still highly significant ($p = 0.004$ and $p < 0.001$, respectively), but also not significant between the intermediate and the elderly group.

| age range | 18-45 y | 46-56 y | 57-80 y |
|---|---|---|---|
| number of subjects | 117 | 114 | 120 |
| fraction for *x* axis | 0.13 | 0.15 | 0.09 |
| fraction for *y* axis | 0.51 | 0.53 | 0.52 |
| fraction for *z* axis | 0.35 | 0.32 | 0.39 |
| SDNN from RRI | 78.5 ms | 65.9 ms | 65.7 ms |
| SDNN from PWI | 83.5 ms | 72.6 ms | 72.3 ms |
| RMSSD from RRI | 54.5 ms | 39.8 ms | 39.5 ms |
| RMSSD from PWI | 71.9 ms | 61.6 ms | 61.7 ms |
| mean PTT | 216.9 ms | 206.7 ms | 200.7 ms |

**Table 4. Comparison of heart rate and HRV parameters from RRI and PWI.** We calculated mean heart rate, SDNN, and RMSSD for (i) all RRI detected in the ECGs, (ii) all RRI associated with PWI (at an accuracy limit of 0.1 s, see Table 2), (iii) all PWI associated with RRI, and (iv) the total set of all detected PWI. Group averages over 351 subjects with detected PWIs are presented.

| | mean heart rate | SDNN | RMSSD |
|---|---|---|---|
| all RRI | 65.1 1/min | 93 ms | 52 ms |
| matched RRI | 64.4 1/min | 70 ms | 45 ms |
| matched PWI | 64.4 1/min | 76 ms | 65 ms |
| all PWI | 64.4 1/min | 115 ms | 138 ms |

the matched PWI closely resemble those for the whole RRI time series. On the other hand, SDNN and in particular RMSSD would be a bit overestimated if no ECGs were available for comparison and these parameters were calculated from all accelerometer-detected PWI (bottom row in Table 4). However, as will be shown below in the results section on the influence of respiration, this cannot be regarded as a problem of our approach, since indeed SDNN and RMSSD are increased by influences of respiratory activity on pulse transit times.

Fig 8 shows Bland-Altman plots as a detailed comparison between the SDNN and RMSSD values derived from RRI and PWI for each subject. Except for six outliers $SDNN_{PWI}$ is larger than $SDNN_{RRI}$ (Fig 8(a)). We can see a slight linear trend in the Bland-Altman plot with a Pearson correlation coefficient of 0.53, since the difference between $SDNN_{PWI}$ and $SDNN_{RRI}$ decreases with higher SDNN. In Fig 8(b) and 8(c) we compared the values of RMSSD, which is a common HRV parameter to estimate parasympatic activity [17]. Furthermore, RMSSD is independent of sleep stages [35]. In nearly all subjects $RMSSD_{PWI}$ is clearly larger than $RMSSD_{RRI}$; the average difference is approximately equal to two standard deviations. This relation holds for both, associated RRI and PWI (Fig 8(b)) and all RRI and PWI (Fig 8(c)), although the difference is much larger in the second case (see also Table 4). The subjects of the three outliers in panel (b) are a subgroup of the six outliers in the SDNN plot (Fig 8(a)). We also see a slight linear trend in the Bland-Altman plot of RMSSD for associated RRI and PWI with a Pearson correlation coefficient of 0.46 (Fig 8(b)), but no clear trend for all RRI and PWI with a Pearson correlation coefficient 0.005 (Fig 8(c)).

The colored symbols in Fig 8, corresponding to the results of the three age groups (see Table 3), show no systematic dependence on age, supporting the conclusion from Fig 7 that our reconstruction of PWI from wrist actigraphy does not depend on age. Furthermore, the mean SDNN and RMSSD values listed in Table 3 for each of the three groups show that the reduction of SDNN and RMSSD with age reported by Schmitt et al. [35] similarly occurs for the HRV parameters derived from RRI and PWI, although their absolute values are different. Apparently, the decrease occurs before the age of approximately 40–50 years, since our results for the last two age groups (46–56 and 57–80 years, respectively) are practically identical.

We note that a recent work also used wrist accelerometry in the frequency range from 4 to 11 Hz to estimate heart rates [36]. However, the study focused on average heart rate (and breathing rate) in intervals of 20 s as determined via spectral analysis, not trying to identify individual heartbeats or beat-to-beat intervals. Besides that, it was limited to 32 h of sleep data from three subjects and 72 minutes of daytime data from twelve subjects. Another recent study determined the average heart rates in 15 subjects using wrist accelerometry [37], reporting an average deviation of 1.6 percent with respect to heart rate from a pulse-oximeter attached to the index finger. This deviation is comparable to the deviation of 0.9 percent we observe between the mean heart rate for all RRI and the PWI-based estimate (Table 4). Another paper from the same group reported that heart rate can be most reliably estimated

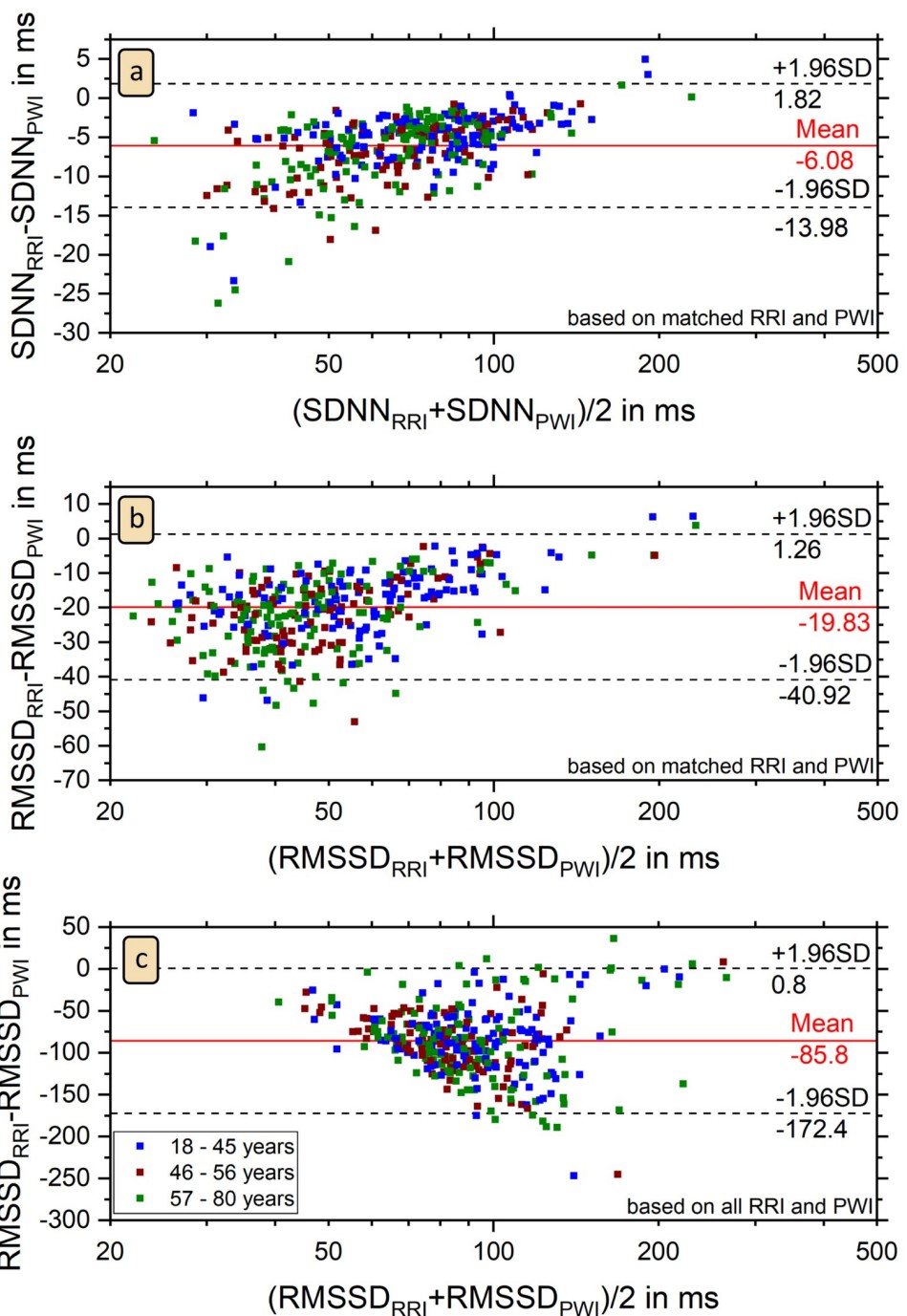

**Fig 8. Bland-Altman plots of SDNN and RMSSD.** The plots show that SDNN and RMSSD values derived from PWI are larger than those derived in the standard way from RRI for nearly all 351 subjects. There are only six outliers for (a) SDNN comparing associated RRI and PWI, three for (b) RMSSD comparing associated RRI and PWI, and eight for (c) RMSSD comparing all RRI and all PWI. In panel (c), one extreme outlier for a subject with 16 percent of ectopic beats and RMSSD$_{all\ RRI}$ = 377 ms, RMSSD$_{all\ PWI}$ = 163 ms does not appear in the plot. No clear differences between the three age groups (see Table 3) can be seen.

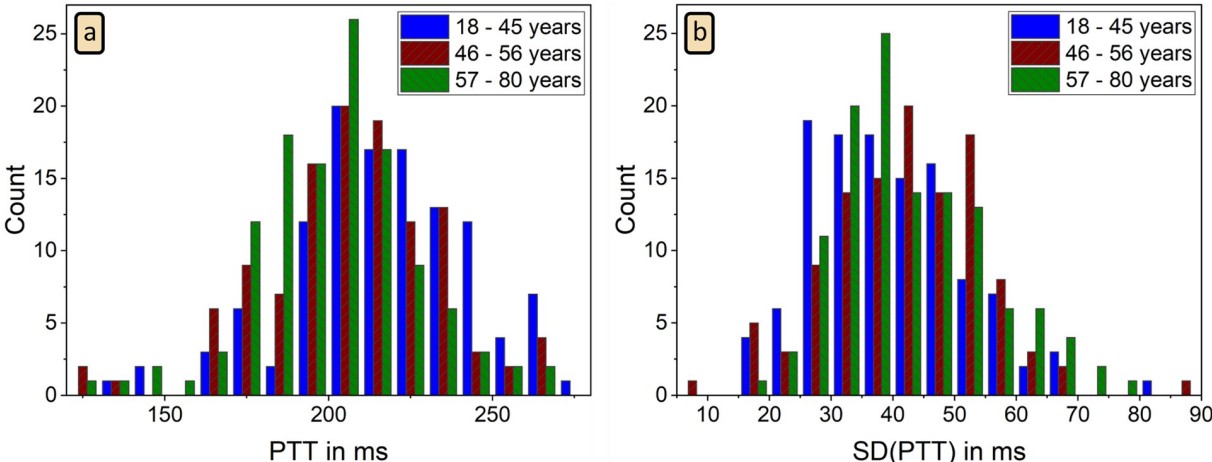

**Fig 9. Histograms of mean PTT and its standard deviation.** Histograms of average PTT defined as the time delay between ECG R peak and the associated PWP, as well as the corresponding standard deviations of the reconstructed PTT intervals. Data of 351 subjects is presented distinguishing three age groups (see Table 3); no PWP were detected in 12 subjects.

via accelerometry, if the sensor is attached to the subjects' upper forearm or the subjects' belly [38].

## Pulse transit times (PTT)

In addition to heartbeat estimation, accelerometer-detected PWP can be used to calculate the time delay between heart beats and PWP, better known as pulse transit time (PTT), if an ECG is simultaneously recorded. The histogram of the mean PTT values in all subjects is shown in Fig 9(a). On average we estimated a PTT of 207 ± 26 ms. This result as well as its range agree with literature [39]. However, PTT values in young subjects seem to be a bit longer than those in the elderly, since a slight difference between the young group and the other two groups can be seen in Fig 9(a) and leads to significantly different means as reported in Table 3. Fig 9(b) shows that the standard deviation of PTT values in each subject (the temporal PTT variation) is distributed around 42 ms and thus comparable with the inter-subject PTT variation.

We think that time series of PTT derived this way could be used in a similar way as ECG-derived RRI are used for studies of HRV, see, e.g., [40]. However, further research will be needed to identify useful PTT-based parameters comparable to the standard HRV parameters. Besides, PTT measurements were suggested to be used as an estimate for continuous blood pressure recording during sleep [41].

## Influence of respiration on PWI

In this subsection, we want to address the reason for the increased values of SDNN and particularly RMSSD as observed when calculating these HRV parameters from acceleration-derived PWI instead of ECG-derived RRI (Tables 3 and 4). It has been known since 1860 that respiration modulates heartbeat frequency, a phenomenon called respiratory sinus arrhythmia (RSA) [42]. A closer look at the tachograms of both RRI and PWI data (Fig 10) clearly shows these periodic oscillations due to RSA. It can also be seen that PWI yield larger variations than RRI suggesting a stronger respiration related modulation.

In order to investigate this observation in greater detail, we compared the power spectra of RRI and PWI time series. We selected all uninterrupted episodes of detected PWI of at least

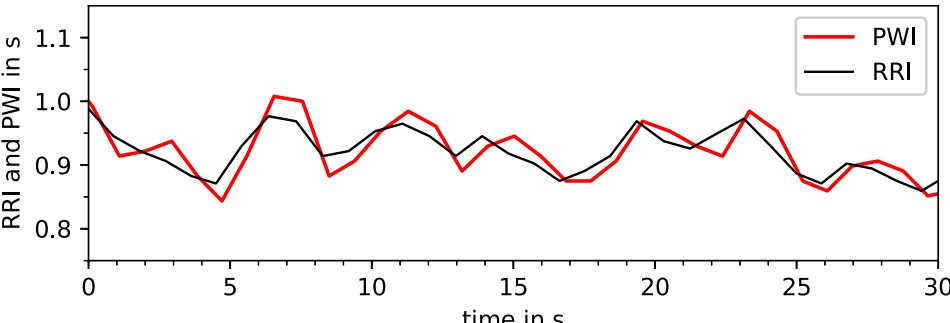

**Fig 10. Tachogram of respiratory sinus arrhythmia.** This plot shows how RRI (black) and PWI (red) follow oscillations due to respiratory sinus arrhythmia.

five minutes duration and calculated the spectra via FFT. Fig 11 shows average power spectra of at least ten five-minute intervals for six typical subjects. Respiration frequency normally lies in the LF band of HRV, between 0.15 and 0.4 Hz [17], see also [43]. The spectra in Fig 11(a) and 11(b) exhibit a high and broad respiratory peak in both, RRI and PWI. The amplitude of the respiratory peak is considerably higher for PWI than for RRI especially in Fig 11(a). But also in Fig 11(c) to 11(e) higher respiratory peaks appear for PWI compared with RRI. Besides this difference the spectra are very similar for both types of intervals. In Fig 11(f) data from a subject with low RSA is presented. We conclude that respiration tends to modulate PWI stronger than RRI.

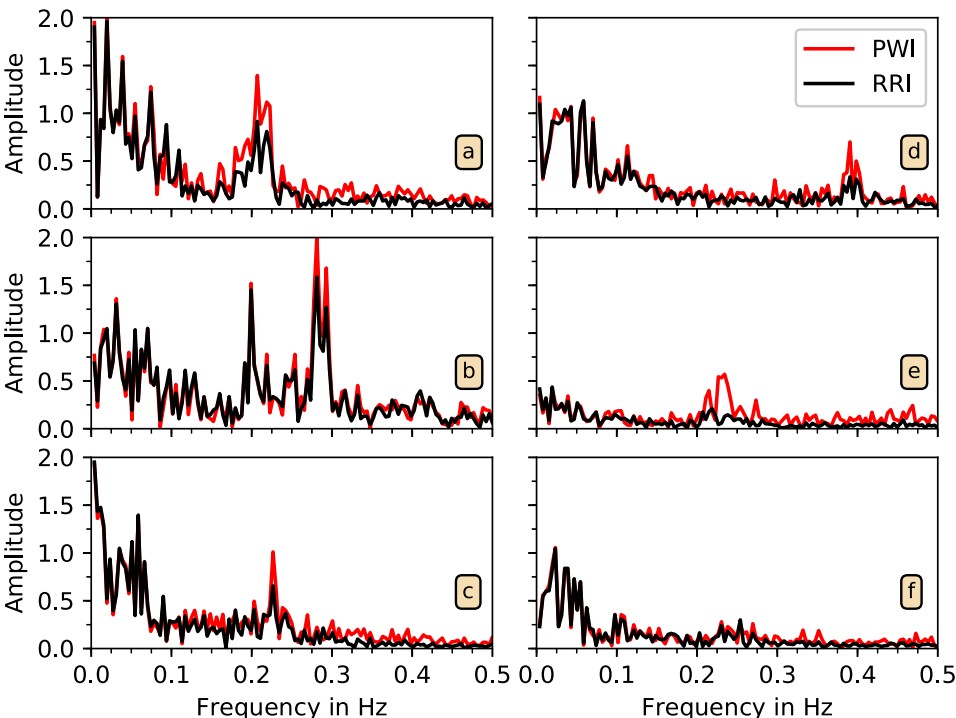

**Fig 11. Spectral analysis of PWI and RRI time series.** The spectra of PWI data (red) and RRI data (black) are shown for six subjects. In all cases except for (f), the peak in the (respiratory) HF band (0.15 to 0.4 Hz) is increased in the PWI-based spectra.

## Conclusion

Although further development, optimization, and validation is necessary, our work represents a novel approach for obtaining long-term nocturnal heartbeat interval time series without the need of ECG recordings (involving electrodes). This could create the possibility to reliably asses heart rate and HRV in large cohort studies solely through accelerometers already used for actigraphy measurements (to characterize activity and sleep patterns). Moreover, our approach could be used to improve plethysmogram-based techniques for measuring heart-beats at the wrist, as currently done in smart watches.

In physiological terms, we show that respiration affects pulse wave velocity in such a way that the respiratory sinus arrhythmia of pulse wave intervals is increased compared with the modulation of RR intervals. However, further research is needed to identify whether the underlying mechanism of increased RSA in PWI is more related to blood pressure modulations or to arterial stiffness modulations.

## Acknowledgments

This study was supported by the German Israel Foundation (GIF) grants I-1298-415.13/2015 and I-1372-303.7/2016 and the German National Cohort study (www.nako.de), funded by the Federal Ministry of Education and Research (BMBF) and the Helmholtz Association. JZ acknowledges support from a Minerva Short-Term Research Grant. We acknowledge the financial support within the funding programme Open Access Publishing by the German Research Foundation (DFG).

## Author Contributions

**Conceptualization:** Jan W. Kantelhardt.

**Data curation:** Alexander Müller.

**Funding acquisition:** Rafael Mikolajczyk, Ronny P. Bartsch, Thomas Penzel.

**Investigation:** Maria Kluge, Luise Pelikan, Antonia Graf.

**Methodology:** Johannes Zschocke.

**Project administration:** Martin Glos.

**Resources:** Martin Glos.

**Software:** Johannes Zschocke.

**Supervision:** Thomas Penzel, Jan W. Kantelhardt.

**Visualization:** Johannes Zschocke.

**Writing – original draft:** Johannes Zschocke, Jan W. Kantelhardt.

**Writing – review & editing:** Ronny P. Bartsch, Jan W. Kantelhardt.

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
