## [Decision Letter · Decision Letter 0]

6 Nov 2019

PONE-D-19-20716

Detection and analysis of pulse waves during sleep via wrist-worn actigraphy

PLOS ONE

Dear Priv.-Doz. Dr. Kantelhardt,

Thank you for submitting your manuscript to PLOS ONE. After careful consideration, we feel that it has merit but does not fully meet PLOS ONE’s publication criteria as it currently stands. Therefore, we invite you to submit a revised version of the manuscript that addresses the points raised during the review process.

Your paper was reviewed by an expert in the field and myself. Although the reviewer's response is somehow positive, there are several points to be clarified. Also, figure legends and discussion need be modified. Please read the comments carefully, and address the issues accordingly. 

We would appreciate receiving your revised manuscript by Dec 21 2019 11:59PM. To enhance the reproducibility of your results, we recommend that if applicable you deposit your laboratory protocols in protocols.io, where a protocol can be assigned its own identifier (DOI) such that it can be cited independently in the future. For instructions see: http://journals.plos.org/plosone/s/submission-guidelines#loc-laboratory-protocols

We look forward to receiving your revised manuscript.

Kind regards,

Tomohiko Ai, M.D., Ph.D.

Academic Editor

PLOS ONE

Journal Requirements:

Reviewers' comments:

Reviewer's Responses to Questions

**Comments to the Author**

1. Is the manuscript technically sound, and do the data support the conclusions?

Reviewer #1: Yes

2. Has the statistical analysis been performed appropriately and rigorously? 

Reviewer #1: Yes

3. Have the authors made all data underlying the findings in their manuscript fully available?

Reviewer #1: Yes

4. Is the manuscript presented in an intelligible fashion and written in standard English?

Reviewer #1: Yes

5. Review Comments to the Author

Reviewer #1: The authors demonstrate the utility of wrist accelerometers to identify heartbeats and to extract heart beat variability. The technical aspects of work are solid, and provided in sufficient detail. The findings are clearly stated, and the manuscript is well-written. The study presents a novel approach to dissect physiological information from wrist activity during sleep and I recommend publication.

There are several important points that the authors should address in the revised manuscript

before publication:

1. Since the age of subjects varies from 18 to 80 years old, having in mind that HRV is reduce in the elderly subjects, do the results depend on the age of the subjects?

2. Many figures (Figs. 1,2,4,5,10) in the manuscript are lack of legends, which creates difficulty for reader to understand the figure. The authors should add legends for these figures.

3. As authors showed in Fig. 5, the x axis (part (a)) yields the most consistent PWP positions with respect to the R peaks in the ECG. Is it always true that in a specific direction (x) that PWP positions are consistent with R peaks, or in the different subjects such consistency changes in different directions?

2. it appears that the authors are not familiar with pioneer studies that utilize wrist actigraphy dynamics. First investigation that demonstrated physiological relevance in the distribution and correlations of wrist activity fluctuations independent of level of physical activity was published in 2003

Hu K, et. al. Novel multiscale regulation in human motor activity. In "Fluctuations and Noise in Biological, Biophysical, and Biomedical Systems", edited by Bezrukov SM, Frauenfelder H, Moss F. SPIE Proceedings 5110, 2003; p. 235-243.

Hu K, et. al. Non-random fluctuations and multi-scale dynamics regulation of human activity. Physica A 2004;337(1-2):307-318.

It was later demonstrated that wrist activity fluctuations are also related to circadian rhythms and to the role of the SCM area in the brain

Ivanov PCh, et. al. Endogenous circadian rhythm in human motor activity uncoupled from circadian influences on cardiac dynamics. Proc. Natl. Acad. Sci. 2007; 104(52):20702-20707.

Hu K, et. al. The suprachiasmatic nucleus functions beyond circadian rhythm generation. Neuroscience 2007; 149(3):508-517.

The authors should give credit to these earlier works.

3. In the context of pause wave interval (PWI) dynamics, the authors should relate their results to a pioneering study in 2002, which utilized PWI to investigate HRV during exercise with the need of ECG electrons:

Karasik R, et. al. Correlation differences in heartbeat fluctuations during rest and exercise. Physical Review E, 2002; 66(6): 062902(4).

4. The authors utilize analytic signal approach in Hilbert transform to extract the amplitude of wrist acceleration signals, and to identify peaks in actigraphy dynamics that correspond to individual heartbeat. The authors should be informed that one of the first publications with Hilbert transform in physiological dynamics was used to detect the amplitude of heart rate variability fluctuations:

Ivanov PCh, et. al. Scaling behaviour of heartbeat intervals obtained by wavelet-based time-series analysis. Nature 1996; 383: 323–327.

Ivanov PCh, et. al. Scaling and universality in heart rate variability distributions. Physica A 1998; 249: 587–593.

6. PLOS authors have the option to publish the peer review history of their article (what does this mean?). If published, this will include your full peer review and any attached files.

Reviewer #1: No

---

## [Author Response · Author response to Decision Letter 0]

22 Nov 2019

(Please see the resubmission letter, where this text is better formated.)

Point-by-point response to the reviewer

We thank the reviewer for his/her careful evaluation and additional comments. Below we outline our point-by-point response to his/her concerns. 

1. Comment: 

Since the age of subjects varies from 18 to 80 years old, having in mind that HRV is reduced in the elderly subjects, do the results depend on the age of the subjects? 

Response: We thank the reviewer for raising this important point. We have divided our database into three age groups of similar sizes, see the new Table 3. As shown by the revised histograms in Fig. 7, the revised Bland-Altman plots (Fig. 8), and Table 3, our algorithm’s performance does not depend on the age of the subjects. We have added one paragraph on page 11 (lines 254-262) to present these findings:

These percentages hardly depend on the age of the subjects. No systematic differences between three age groups of approximately equal size (see Table 3) can be observed when comparing the corresponding histograms for each color in Fig 7. This indicates that the reconstruction of pulse waves from wrist actigraphy as presented in this paper does not depend on age. Furthermore, the results in Table 3 show that there is no systematic age dependence in the PWI algorithm selection of particular orientation axes. Across all age groups, the y axis acceleration data is selected most frequently for the PWI detection, followed by the z axis data. The x axis, pointing towards the hand, is only quite rarely selected.

However, we clearly see the reduction with age in HRV as quantified by SDNN and RMSSD. Interestingly, from the RR interval analysis we obtain very similar values of SDNN and RMSSD for the age groups as have been found in Schmitt et al. IEEE 2009;56(5):1564-1573 (reference [35] in the paper). To better explain this point, we have modified the caption of Fig. 8 and added the following text on page 12 (lines 291-299):

The colored symbols in Fig 8, corresponding to the results of the three age groups (see Table 3), show no systematic dependence on age, supporting the conclusion from Fig 7 that our reconstruction of PWI from wrist actigraphy does not depend on age. Furthermore, the mean SDNN and RMSSD values listed in Table 3 for each of the three groups show that the reduction of SDNN and RMSSD with age reported by Schmitt et al. [35] similarly occurs for the HRV parameters derived from RRI and PWI, although their absolute values are different. Apparently, the decrease occurs before the age of approximately 40-50 years, since our results for the last two age groups (46-56 and 57-80 years, respectively) are practically identical. 

We have also revised Fig. 9 to show the age dependence of pulse transit times and added the following text on page 13 (lines 317-320):

However, PTT values in young subjects seem to be a bit longer than those in the elderly, since a slight difference between the young group and the other two groups can be seen in Fig 9(a) and leads to significantly different means as reported in Table 3. 

2. Comment:

Many figures (Figs. 1,2,4,5,10) in the manuscript are lack of legends, which creates difficulty for reader to understand the figure. The authors should add legends for these figures. 

Response: We agree with the reviewer and have added the legends accordingly.

3. Comment:

As authors showed in Fig. 5, the x axis (part (a)) yields the most consistent PWP positions with respect to the R peaks in the ECG. Is it always true that in a specific direction (x) that PWP positions are consistent with R peaks, or in the different subjects such consistency changes in different directions? 

Response: We have addressed this important point in our new Table 3, where the fractions of pulse wave peaks derived from the data of each acceleration axis are reported for all three age groups. The results show that PWP from the x axis are in fact least frequent; more than half of the consistent peaks come from the y axis. We have added the following two sentences (lines 258-262).

Furthermore, the results in Table 3 show that there is no systematic age dependence in the PWI algorithm selection of particular orientation axes. Across all age groups, the y axis acceleration data is selected most frequently for the PWI detection, followed by the z axis data. The x axis, pointing towards the hand, is only quite rarely selected.

4. Comment:

It appears that the authors are not familiar with pioneer studies that utilize wrist actigraphy dynamics. First investigation that demonstrated physiological relevance in the distribution and correlations of wrist activity fluctuations independent of level of physical activity was published in 2003: Hu K, et. al. Novel multiscale regulation in human motor activity. In "Fluctuations and Noise in Biological, Biophysical, and Biomedical Systems", edited by Bezrukov SM, Frauenfelder H, Moss F. SPIE Proceedings 5110, 2003; p. 235-243; Hu K, et. al. Non-random fluctuations and multi-scale dynamics regulation of human activity. Physica A 2004;337(1- 2):307-318. 

It was later demonstrated that wrist activity fluctuations are also related to circadian rhythms and to the role of the SCM area in the brain: Ivanov PCh, et. al. Endogenous circadian rhythm in human motor activity uncoupled from circadian influences on cardiac dynamics. Proc. Natl. Acad. Sci. 2007; 104(52):20702-20707. Hu K, et. al. The suprachiasmatic nucleus functions beyond circadian rhythm generation. Neuroscience 2007; 149(3):508-517. 

The authors should give credit to these earlier works. 

Response: We thank the reviewer for pointing this out. We have now added these references accordingly on pages 2 and 3 in the revised manuscript. The new text reads:

First investigations that demonstrated physiological relevance in the distribution and autocorrelations of wrist activity fluctuations independent of level of physical activity were published by Hu et al. [13,14]. In later studies it has been shown that wrist activity fluctuations are also related to the circadian rhythm and to the role of the suprachiasmatic nucleus in the brain [15,16] that is responsible for regulating many different body functions on a 24-hour cycle.

5. Comment:

In the context of pause wave interval (PWI) dynamics, the authors should relate their results to a pioneering study in 2002, which utilized PWI to investigate HRV during exercise with the need of ECG electrodes: Karasik R, et. al. Correlation differences in heartbeat fluctuations during rest and exercise. Physical Review E, 2002; 66(6): 062902(4). 

Response: We thank the reviewer for pointing this out. We have added this reference on page 4 in the revised manuscript together with the following text: 

We refer to [22] for an early application of using pulse wave intervals from plethysmography to study cardiac dynamics and investigate HRV during rest and exercise without ECG electrodes. 

6. Comment:

The authors utilize analytic signal approach in Hilbert transform to extract the amplitude of wrist acceleration signals, and to identify peaks in actigraphy dynamics that correspond to individual heartbeat. The authors should be informed that one of the first publications with Hilbert transform in physiological dynamics was used to detect the amplitude of heart rate variability fluctuations: Ivanov PCh, et. al. Scaling behaviour of heartbeat intervals obtained by wavelet-based time-series analysis. Nature 1996; 383: 323–327; Ivanov PCh, et. al. Scaling and universality in heart rate variability distributions. Physica A 1998; 249: 587–593. 

Response: We thank the reviewer for pointing this out. These references are now added on page 8 with the following text:

Among the first applications of this approach to physiological dynamics are the works of Ivanov et al. who used Hilbert transform to detect the amplitude of heart rate variability fluctuations [32,33].

---

## [Editor Report · Decision Letter 1]

9 Dec 2019

Detection and analysis of pulse waves during sleep via wrist-worn actigraphy

PONE-D-19-20716R1

Dear Dr. Kantelhardt,

We are pleased to inform you that your manuscript has been judged scientifically suitable for publication and will be formally accepted for publication once it complies with all outstanding technical requirements.

With kind regards,

Tomohiko Ai, M.D., Ph.D.

Academic Editor

PLOS ONE
---

## [Editor Report · Acceptance letter]

18 Dec 2019

PONE-D-19-20716R1 

Detection and analysis of pulse waves during sleep via wrist-worn actigraphy 

Dear Dr. Kantelhardt:

I am pleased to inform you that your manuscript has been deemed suitable for publication in PLOS ONE. Congratulations! Your manuscript is now with our production department. 

With kind regards,

on behalf of

Dr. Tomohiko Ai 

Academic Editor

PLOS ONE